# Retinoic Acid Receptor Alpha (RARα) in Macrophages Protects from Diet-Induced Atherosclerosis in Mice

**DOI:** 10.3390/cells11203186

**Published:** 2022-10-11

**Authors:** Fathima N. Cassim Bawa, Raja Gopoju, Yanyong Xu, Shuwei Hu, Yingdong Zhu, Shaoru Chen, Kavita Jadhav, Yanqiao Zhang

**Affiliations:** 1Department of Integrative Medical Sciences, Northeast Ohio Medical University, Rootstown, OH 44272, USA; 2School of Biomedical Sciences, Kent State University, Kent, OH 44240, USA

**Keywords:** all-trans retinoic acid, atherosclerosis, retinoic acid receptor alpha, cholesterol efflux, macrophages

## Abstract

Retinoic acid signaling plays an important role in regulating lipid metabolism and inflammation. However, the role of retinoic acid receptor alpha (RARα) in atherosclerosis remains to be determined. In the current study, we investigated the role of macrophage RARα in the development of atherosclerosis. Macrophages isolated from myeloid-specific *Rarα^-/-^* (*Rarα^Mac-/-^*) mice showed increased lipid accumulation and inflammation and reduced cholesterol efflux compared to *Rarα^fl/fl^* (control) mice. All-trans retinoic acid (AtRA) induced ATP-binding cassette subfamily A member 1 (*Abca1*) and *Abcg1* expression and cholesterol efflux in both *Rarα^Mac-/-^* mice and *Rarα^fl/fl^* mice. In *Ldlr^-/-^* mice, myeloid ablation of RARα significantly reduced macrophage *Abca1* and *Abcg1* expression and cholesterol efflux, induced inflammatory genes, and aggravated Western diet-induced atherosclerosis. Our data demonstrate that macrophage RARα protects against atherosclerosis, likely via inducing cholesterol efflux and inhibiting inflammation.

## 1. Introduction

Atherosclerosis is a chronic inflammatory cardiovascular disease (CVD) characterized by fatty lesions in the arterial wall and is responsible for most cardiovascular morbidity and mortality in the world [1]. Macrophages play a critical role in atherosclerosis via foam cell formation and inflammatory responses [2]. Cholesterol efflux to lipid-poor apolipoproteins or high-density lipoproteins (HDL) occurs mainly via ATP-binding cassette subfamily A member 1 (ABCA1) and ATP- binding cassette subfamily G member 1 (ABCG1). Cholesterol efflux is the first step in reverse cholesterol transport (RCT), a metabolic pathway whereby excess cholesterol in peripheral tissues is removed and transported to the liver [3,4]. Cholesterol efflux from macrophages is likely the most important step in RCT, as the impairment of macrophage RCT may cause foam cell formation and inflammatory response, leading to the development of atherosclerosis [5].

Vitamin A is often studied for its role in metabolic diseases, including cardiovascular disease (CVD). A vitamin-A-deficient diet is shown to accelerate atherosclerosis in *Apoe*^-/-^ mice, while dietary supplementation of vitamin A prevents atherosclerosis [6]. Vitamin A exerts its function through active metabolites—all-trans retinoic acid (AtRA) and 9-cis retinoic acid (9-cis-RA), which activate nuclear receptors retinoic acid receptors (RARα, RARβ, and RARγ) and retinoid-X-receptors (RXRα, RXRβ, and RXRγ), respectively. RARα heterodimerizes with RXR and binds to DNA motifs known as retinoic acid response elements (RAREs) and regulates gene transcription upon interaction with its ligands, resulting in activation or repression of genes involved in cellular growth, differentiation, and metabolism [7,8,9]. RARα is expressed in many cells and has been extensively investigated in promyelocytic leukemia [10].

Liver X receptor (LXR) transcriptionally regulates gene expression by binding as heterodimers with RXR to the LXR-response elements (LXRE)—the direct repeats spaced by four base pairs (DR4). LXR is known to induce the promoter activity and expression of ABCA1 and ABCG1 [11,12]. AtRA induces ABCA1 and ABCG1 expression by binding to the DR4 element in the promoter region of ABCA1 or ABCG1 in human primary monocytes/macrophages [13,14]. As a result, AtRA treatment enhances cholesterol efflux, which may also involve the activation of LXR [15]. Consistent with these findings, AtRA inhibits the diet-induced development of atherosclerosis in animal models [16,17,18,19]. Nonetheless, it is unclear which receptor(s) mediate the effect of AtRA on cholesterol efflux. 

The role of RARα in atherosclerosis has not been investigated before. Furthermore, it is unclear whether AtRA induces macrophage cholesterol efflux via RARα. In this report, we show that loss of RARα reduces macrophage cholesterol efflux, and AtRA induces macrophage cholesterol efflux independent of RARα. In addition, myeloid ablation of RARα in *Ldlr*^-/-^ mice reduces ABCA1 and ABCG1 expression and cholesterol efflux in macrophages and aggravates Western diet-induced atherosclerosis.

## 2. Materials and Methods

### 2.1. Mice

*Rarα^fl/fl^* mice on a C57BL/6 background were generously gifted by Dr. Yasmine Balkaid (NIH/NIAID) and have been previously described by Dr. Chambon and colleagues [20]. *Ldlr^-/-^* mice and LysM-Cre mice were purchased from the Jackson Laboratory and had a C57BL/6 background. These mice were cross-bred to generate macrophage-specific *Rarα^fl/fl^* (control), *Rarα*^-/-^ (*Rarα^Mac-/-^*), *Rarα^fl/fl^Ldlr^-/-^,* and *Rarα^Mac-/-^Ldlr^-/-^* mice. The Western diet (42% kcal from fat/0.2% cholesterol) was purchased from Envigo (Cat# TD.88137). For the feeding studies, 8-week-old male mice were randomly allocated and given the Western diet for 16 weeks. Unless otherwise stated, all mice were fasted for 5–6 h prior to euthanasia. All the animal studies were approved by the Institutional Animal Care and Use Committee at Northeast Ohio Medical University.

### 2.2. Primary Peritoneal Macrophage Isolation and Cell Culture

Primary peritoneal macrophages were isolated, as described previously [21]. Briefly, mice were injected i.p. with sterile 4% Brewer’s thioglycolate medium. After 4 days, peritoneal macrophages were isolated from the peritoneal cavity using cold phosphate-buffered saline (PBS). Macrophages were separated after centrifugation and resuspended in cell culture medium. Mouse RAW267.4 cells were purchased from the American Type Culture Collection. RAW267.4 cells or peritoneal macrophages were cultured in DMEM containing 10% FBS. In some studies, they were treated with acetylated LDL (Ac-LDL) (25 μg/mL; Kalen Biomedical, Montgomery Village, MD, USA) to induce lipid accumulation and/or treated with 2 μM AtRA—a pan agonist for RARs (Sigma Aldrich, St. Louis, MO, USA) or AM580—a specific agonist for RARα (Sigma Aldrich, St. Louis, MO, USA). 

### 2.3. Quantitative Real-Time PCR

Cells were collected in TRIzol reagent (Invitrogen, Carlsbad, CA, USA) after washing with PBS. About 100 mg tissues were homogenized in TRIzol reagent. Chloroform was added, followed by vortex for 10–15 s and centrifugation at 12,000× *g* rpm for 15 min. The supernatant was collected, and equal volume of isopropanol was added and centrifuged at 12,000× *g* rpm for 20 min to the precipitate RNA. After washing with 70% ethanol, RNA pellet was air dried for 5–10 min and then dissolved in nuclease-free water. The cDNA synthesis was performed using the TaqMan Reverse Transcription Kit (Thermo Fisher Scientific, Carlsbad, CA, USA) from total RNA according to the manufacturer’s instructions. Briefly, samples were incubated at 25 °C for 10 min, 48 °C for 30 min, and then 95 °C for 5 min using a thermal cycler program. The cDNA was diluted further, and mRNA levels were quantified by quantitative real-time PCR (qRT-PCR) using PowerUp SYBR Green Master Mix (Thermo Fisher Scientific, Austin, TX, USA) on a 7500 Real Time PCR machine (Applied Biosystems, Foster City, CA, USA). Conditions for the qPCR reaction were as follows: 50 °C for 2 min, 95 °C for 10 min, 40 cycles of 95 °C for 15 s, and 60 °C for 1 min. Relative mRNA levels were quantified using the 2^−ΔΔCT^ method and normalized to *36b4*. The primer sequences for qRT-PCR are presented in Appendix A.

### 2.4. Western Blotting

Cells were collected into ice-cold protein lysis buffer containing protease inhibitors after washing with sterile PBS. Lysates were rotated at 4 °C and then centrifuged at 15,000× *g* rpm for 30 min. Supernatant was collected, and protein concentration was measured using Pierce BCA protein assay (Thermo Fisher Scientific, Rockford, IL, USA). The protein concentration was normalized with 6X Laemmli SDS sample buffer (Alfa Aesar, Tewksbury, MA, USA) and water and heated at 95 °C for 10 min. For detecting ABCG1 and alpha-tubulin, the samples were loaded into an 8% SDS-PAGE gel and transferred using the semi-dry transfer method (Bio-Rad Trans Blot Turbo Transfer System). For detecting ABCA1 proteins, the protein samples were incubated at room temperature for 20 min, loaded into a 4.8% SDS-PAGE gel, run overnight at 4 °C, and transferred onto Immobilon-P PVDF Membrane (Millipore Sigma, Burlington, MA, USA). All the transferred membranes were blocked in 5% milk in TBST (TBS, 0.1% *v*/*v* Tween 20) for at least 1 h. The membrane was then incubated at 4 °C overnight with a primary antibody (1:1000) in 5% milk in TBST. Next day, the membrane was washed three times using TBST (TBS with 0.1% *v*/*v* Tween 20) and incubated with a secondary antibody (1:3000 dilution) in 5% milk in TBST for 1 h. All the secondary antibodies were conjugated with horseradish peroxidase for chemiluminescent detection. Blots were developed using SuperSignal West Pico Chemiluminescent Substrate (Thermo Fisher Scientific, Rockford, IL, USA), and the proteins were visualized and imaged using Amershar Imager 680 System. Alpha-tubulin was used as a loading control. Antibodies against ABCA1 (NB400-105) and ABCG1 (NB400-132) were purchased from Novus, Centennial, CO, USA. Alpha-tubulin antibody was purchased from Abcam, Waltham, MA, USA (ab4074). 

### 2.5. Cholesterol Efflux

Cholesterol efflux was performed as described previously [22]. In brief, peritoneal macrophages were seeded in a 24-well plate and loaded with 0.5 mCi/mL [^3^H] cholesterol (PerkinElmer, Waltham, MA, USA) for 24 h in the presence or absence of Ac-LDL (50 mg/mL). After extensive wash with PBS, cells were incubated in DMEM supplemented with 0.2% fatty acid-free BSA for 4 h. The cells were washed again with PBS and then incubated in fresh DMEM containing 0.2% BSA in the presence or absence of acceptors ApoA-I (15 mg/mL) or HDL (50 mg/mL). Supernatants were collected after 4 h, and the level of [^3^H] cholesterol was determined by scintillation counting. Values were expressed as a percentage of total [^3^H] cholesterol content (effluxed plus cellular [^3^H] cholesterol).

### 2.6. Analysis of Plasma AST, ALT, and Lipids

Plasma triglycerides (TG), cholesterol, glucose, AST, and ALT levels were determined using Infinity reagents (Thermo Fisher Scientific, Middletown, VA, USA). Non-esterified free fatty acids (NEFA) were determined using a kit from Wako Chemicals, Richmond, VA, USA.

### 2.7. Analysis of Intracellular and Hepatic Lipids

Mouse primary peritoneal macrophages or 100 mg liver were homogenized in methanol, and lipids were extracted in chloroform/methanol (2:1 *v*/*v*) as described previously [23]. Intracellular and hepatic TG and total cholesterol (TC) levels were quantified using Infinity reagents (Thermo Fisher Scientific, Middletown, VA, USA). Free cholesterol (FC) and NEFA were determined using a kit from Wako Chemicals, Richmond, VA, USA.

### 2.8. Analysis of Hepatic Hydroxyproline

About 40–50 mg liver was homogenized in the distilled water, and hydroxyproline level in liver was quantified using a kit from Cell BioLabs (cat# STA675).

### 2.9. Histological Analysis of Aortic Lesions

The heart was isolated from PBS-perfused mice, fixed in 10% formalin, and embedded in optical cutting temperature compound (OCT). The aorta, including the ascending arch, thoracic, and abdominal segments, and aortic root, were isolated and gently cleaned of the adventitia. Sectioned aortic roots or en face aortas were stained with oil red O, and the atherosclerotic plaque size was determined using the Image-Pro software (Media Cybernetics, Rockville, MD, USA) as described previously [24].

### 2.10. Hematoxylin and Eosin (H&E) or Picrosirius Red Staining

Liver tissues were fixed in 10% formalin and then embedded in OCT or paraffin. Liver sections were stained with H&E or Picrosirius red. Images were acquired using an Olympus microscope.

### 2.11. Body Composition Analysis 

Mouse body fat mass was measured by EchoMRI-700 (Echo Medical Systems, Houstan, TX, USA).

### 2.12. Statistical Analysis

All data were expressed as mean ± SEM. Statistical significance was analyzed using unpaired student *t*-test or analysis of variance (ANOVA; for more than two groups) (GraphPad Prism 9.4.1, San Diego, CA, USA). Differences were considered statistically significant at *p* < 0.05.

## 3. Results

### 3.1. Activation of RAR Signaling Attenuates Lipid Accumulation in RAW264.7 Cells

To investigate whether activation of the RAR signaling affected lipid homeostasis in macrophages, we treated RAW267.4 cells with RAR agonists AtRA and AM580 (a selective RARα agonist) in the presence of vehicle or Ac-LDL. AtRA and AM580 significantly reduced cellular levels of triglycerides (TG) (Figure 1A), non-esterified free fatty acids (NEFA) (Figure 1B), total cholesterol (TC) (Figure 1C), free cholesterol (FC) (Figure 1D) and cholesterol esters (CE) (Figure 1E) in the presence or absence of Ac-LDL. AtRA and AM580 also significantly induced *Abca1* (Figure 1F) and *Abcg1* (Figure 1G) expression in the presence of Ac-LDL. These results indicate that activation of the RAR signaling attenuates lipid accumulation in macrophages. 

### 3.2. Loss of RARα in Macrophages Increases Intracellular Lipid Accumulation and Inflammation

We have previously shown that loss of RARα causes accumulation of TG, NEFA, and cholesterol in hepatocytes [25]. To determine whether loss of RARα in macrophages had similar effects, we isolated peritoneal macrophages from *Rarα^fl/fl^* mice and *Rarα^Mac-/-^* mice and treated them with Ac-LDL. Loss of RARα led to the accumulation of TG, NEFA, TC, and FC in macrophages (Figure 2A), supporting the role of RAR activation in macrophage lipid homeostasis (Figure 1). Interestingly, AtRA-induced gene expression of *Abca1*, *Abcg1,* and repressed tumor necrosis factor α (*Tnfα*) and interleukin β (*Il1β*) expression in macrophages isolated from both *Rarα^fl/fl^* mice and *Rarα^Mac-/-^* mice (Figure 2B–E). In contrast, AM580 regulated these genes in macrophages isolated from *Rarα^fl/fl^* mice but not in *Rarα^Mac-/-^* mice (Figure 2B–E). These data suggest that AM580, but not AtRA, is a selective agonist for RARα. 

We also isolated peritoneal macrophages from the Western diet (WD)-fed *Rarα^fl/fl^* mice and *Rarα^Mac-/-^* mice. No changes between the two groups were observed in body weight, fat content, plasma levels of transaminases, TG, NEFA or TC, liver to body weight ratio, or hepatic levels of TG, NEFA, cholesterol or hydroxyproline, or liver morphology (Appendix A). However, RARα-deficient macrophages had higher levels of TG, NEFA, TC, and CE (Figure 2F), and had lower levels of *Abca1* and *Abcg1* but higher levels of *Tnfα* and *Il1β* (Figure 2G). Thus, the data in Figure 2 indicate that loss of RARα increases macrophage lipid accumulation in vitro and in vivo. 

### 3.3. AtRA and RARα Promote Macrophage Cholesterol Efflux

The data of Figure 1 and Figure 2 show that both AtRA and RARα regulate the mRNA levels of ABCA1 and ABCG1, two key transporters responsible for cholesterol efflux from macrophages [26]. AtRA significantly induced ABCA1 and ABCG1 protein levels in acetylated LDL (Ac-LDL)-treated macrophages isolated from both *Rarα^fl/fl^* and *Rarα^Mac-/-^* mice (Figure 3A–C). Interestingly, RARα deficiency reduced ABCG1, but not ABCA1, protein levels by 71% (Figure 3A–C). ABCG1 is required for cholesterol efflux to HDL, whereas ABCA1 is required for cholesterol efflux to ApoA-I [27]. Therefore, we performed cholesterol efflux using HDL as an acceptor. AtRA increased cholesterol efflux to HDL from macrophages isolated from both *Rarα^fl/fl^* and *Rarα^Mac-/-^* mice in the absence (Figure 3D) or presence (Figure 3E) of Ac-LDL, whereas RARα deficiency significantly reduced macrophage cholesterol efflux to HDL in the absence or presence of Ac-LDL (Figure 3D,E). Thus, the data of Figure 3 indicate that RARα deficiency inhibits macrophage cholesterol efflux, and AtRA promotes macrophage cholesterol efflux independent of RARα. 

### 3.4. Macrophage RARα Deficiency in Ldlr^-/-^ Mice Promotes Foam Cell Formation by Inhibiting Cholesterol Efflux to HDL and ApoA-I

To investigate whether macrophage RARα played a role in atherosclerosis, we generated *Rarα^Mac-/-^Ldlr*^-/-^ double knockout mice and their control (*Rarα^Mac+/+^Ldlr*^--/-^) mice and fed them with a Western diet (WD) for 16 weeks. There were no changes in plasma levels of transaminases, glucose, TG, NEFA, HDL-C, or LDL-C, or hepatic levels of TG, NEFA, or cholesterol between the two genotypes (Appendix A). However, peritoneal macrophages isolated from *Rarα^Mac-/-^Ldlr*^-/-^ mice accumulated more TG, NEFA, TC, FC, and CE (Figure 4A). Consistent with increased lipid accumulation, macrophages isolated from *Rarα^Mac-/-^Ldlr*^-/-^ mice had 43% and 89% reductions in ABCA1 (Figure 4B) and ABCG1 (Figure 4C) protein levels, respectively. In line with the latter data, macrophages isolated from *Rarα^Mac-/-^Ldlr*^-/-^ mice had significant reductions in cholesterol efflux to HDL (Figure 4D) or ApoA-I (Figure 4E). These data suggest that RARα deficiency can promote foam cell formation in hyperlipidemic *Ldlr*^-/-^ mice by inhibiting cholesterol efflux to both HDL and ApoA-I. 

### 3.5. Macrophage RARα Deficiency in Ldlr^-/-^ Mice Promotes the Development of Atherosclerosis

In addition to inhibiting macrophage cholesterol efflux, RARα deficiency in *Ldlr*^-/-^ mice also induced *Tnfα* and *Il1β* expression in macrophages (Figure 5A). Therefore, *Rarα^Mac-/-^Ldlr*^-/-^ mice had a 50% increase in en face aortic lesions (Figure 5B,C), and a 34% increase in aortic root lesions (Figure 5D,E). The changes in atherosclerotic lesions were not associated with any changes in plasma TG or cholesterol levels (Appendix A). Thus, macrophage RAR*α* deficiency promotes the development of atherosclerosis, likely via inhibition of macrophage cholesterol efflux and induction of macrophage inflammation. 

## 4. Discussion

So far, no genetic RARα models have been used for studying the role of RARα signaling in atherosclerosis. In this work, we show that macrophage-specific RARα plays a key role in regulating macrophage cholesterol efflux, inflammation, and atherosclerosis. RARα is required for ABCG1 expression with a much less impact on ABCA1 expression. ABCG1 and ABCA1 are required from macrophage cholesterol efflux to HDL and ApoA-I, respectively [27]. Nonetheless, macrophage RARα deficiency prevents cholesterol efflux to both HDL and ApoA-I in *Ldlr*^-/-^ mice. Since macrophage cholesterol efflux is the first step in RCT and is critical in preventing atherogenesis, macrophage RARα deficiency promotes the development of atherosclerosis likely through, at least in part, inhibiting cholesterol efflux. The increased macrophage inflammatory response may play a role in the progression of atherosclerosis in RARα-deficient mice. 

AtRA, a derivative of vitamin A, functions through binding to RARs. AtRA has been shown to inhibit atherosclerosis in high-fat diet-fed *Apoe*^-/-^ mice [28] or rabbits [16,18]. However, it remains to be determined whether AtRA inhibits atherogenesis via RARα, RARβ, RARγ, or other receptor(s). By using *Rarα^Mac-/-^* macrophages, our data show that AtRA induces macrophage cholesterol efflux and inhibits macrophage inflammation independent of RARα. In contrast, our data show that AM580 regulates cholesterol efflux dependent on RARα. AM580 is a RARα-specific agonist that is already proven to protect from diet-induced hyperlipidemia and hepatic lipid accumulation by inhibiting Apolipoprotein C-III (ApoC-III) [29,30]. Thus, it will be interesting to investigate whether AM580 inhibits the development of atherosclerosis by activation of RARα. 

Previous studies have shown that RAR signaling regulates ABCA1 and ABCG1 expression by directly binding to the DR4 elements in their promoters [13,29]. Macrophages, unlike the liver and intestine, do not contribute to plasma lipid levels. As a result, we do not see any change in plasma lipid levels in mice lacking macrophage RARα. It remains unclear how RARα signaling inhibits macrophage inflammatory response. RAR*α*-deficient macrophages accumulate more FFAs and FC, which may trigger an inflammatory response. We have previously shown that RARα inhibits fatty acid uptake and triglyceride synthesis in the hepatocytes [25]. RAR*α*-deficient macrophages accumulate more FFAs and TG, likely via inducing fatty acid uptake and TG synthesis. 

Apart from hepatocytes and macrophages, RAR*α* also plays an important role in metabolic homeostasis in adipocytes. Activation of RARs by AtRA promotes perivascular adipose tissue browning and adiponectin production in *Apoe*^-/-^ mice [28]. While adipocyte-specific expression of dominant-negative RAR*α* causes glucose intolerance and hepatic steatosis in mice [31]. Nonetheless, how adipocyte RAR*α* regulates lipid metabolism is not well understood.

One of the limitations of this study is that we only used male mice largely because male mice are more prone to develop metabolic disorders [32] and atherosclerotic plaques [33]. However, our in vitro studies, in which macrophages were cultured for 48 h in the absence of hormones, support our in vivo findings, suggesting that the results from both male and female mice will likely be similar. Nonetheless, further studies remain needed to test whether RARα functions similarly in both genders.

In summary, we have identified a novel role of macrophage RARα in regulating cholesterol efflux, inflammation, and atherosclerosis. RAR*α* in hepatocytes is also shown to protect against liver steatosis [25]. Thus, targeting RARα may represent a novel and promising strategy for the treatment of atherosclerosis.

## Figures and Tables

**Figure 1 cells-11-03186-f001:**
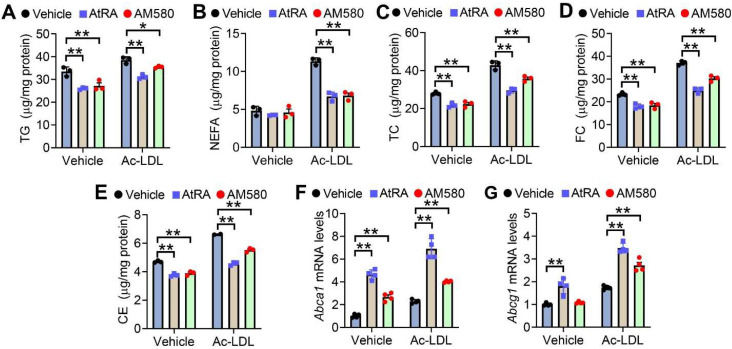
Activation of RAR signaling attenuates lipid accumulation in RAW246.7 cells. (**A**–**E**) Raw246.7 cells were treated with vehicle or acetylated LDL (Ac-LDL, 25 μg/mL), AtRA (2 μM), or AM580 (2 μM) for 48 h (*n* = 3). Intracellular levels of triglycerides (TG) (**A**), non-esterified fatty acids (NEFA) (**B**), total cholesterol (TC) (**C**), free cholesterol (FC) (**D**), and cholesterol esters (CE) (**E**) were determined. (**F**,**G**) Raw246.7 cells were treated with vehicle or Ac-LDL (25 μg/mL), AtRA (2 μM), or AM580 (2 μM) for 24 h (*n* = 4). *Abca1* (**F**) or *Abcg1* (**G**) mRNA levels were determined. In (**A**–**G**), ordinary two-way ANOVA with Tukey’s multiple comparisons test was used for statistical analysis. All the data are expressed as mean ± SEM. * *p* < 0.05, ** *p* < 0.01.

**Figure 2 cells-11-03186-f002:**
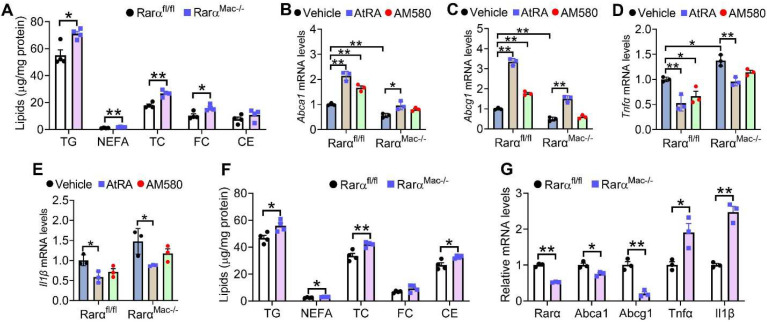
Loss of RARα induces lipid accumulation and inflammation in macrophages. (**A**–**E**) Peritoneal macrophages were isolated from *Rar**α**^fl/fl^* mice or *Rar**α**^Mac-/-^* mice, and then treated with Ac-LDL (25 mg/mL) for 48 h. The cells were also treated with vehicle, AtRA (2 μM), or AM580 (2 μM). Intracellular lipids (**A**) and mRNA levels of *Abca1* (**B**), *Abcg1* (**C**), *Tnf**α* (**D**) or *Il1β* (**E**) were determined (*n* = 4). (**F**,**G**) *Rar**α**^fl/fl^* mice and *Rar**α**^Mac-/-^* mice were fed a Western diet for 16 weeks. Intracellular lipid (**F**) and mRNA levels (**G**) were determined (*n* = 3). In (**A**,**F**,**G**), unpaired two-tailed Students’ *t*-test was used for statistical analysis. In B-E, ordinary two-way ANOVA with Tukey’s multiple comparisons test was used for statistical analysis. All the data are expressed as mean ± SEM. * *p* < 0.05, ** *p* < 0.01.

**Figure 3 cells-11-03186-f003:**
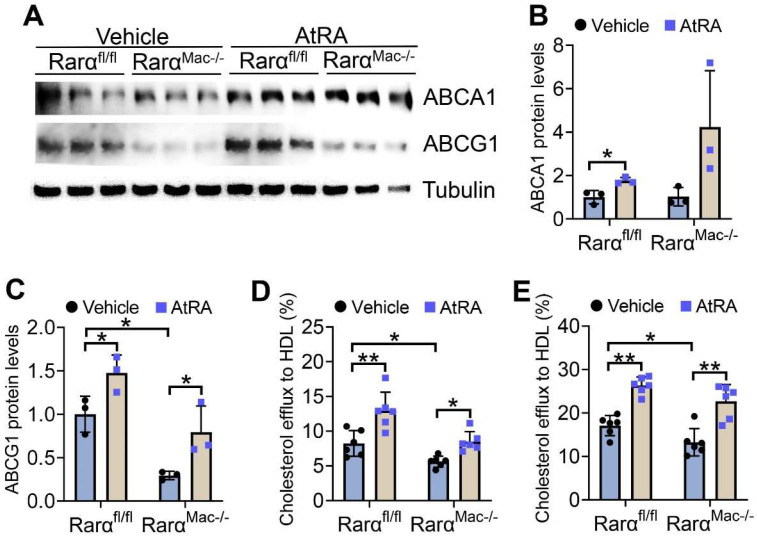
AtRA and RARα promote macrophage cholesterol efflux. (**A**–**C**) Peritoneal macrophages were isolated from *Rar**α**^fl/fl^* mice or *Rar**α**^Mac-/-^* mice, and then treated with Ac-LDL (25 mg/mL) for 48 h. The cells were also treated with vehicle or AtRA (2 μM). Western blot assays were performed (**A**), and ABCA1 (**B**) or ABCG1 (**C**) protein levels were quantified (*n* = 3). (**D**,**E**) Peritoneal macrophages were isolated from *Rar**α**^fl/fl^* mice or *Rar**α**^Mac-/-^* mice, and then treated with vehicle (**D**) or Ac-LDL (25 mg/mL) (**E**) for 48 h. The cells were also treated with vehicle or AtRA (2 μM). Cholesterol efflux to HDL was performed (*n* = 6). In (**B**–**E**), ordinary two-way ANOVA with Tukey’s multiple comparisons test was used for statistical analysis. All the data are expressed as mean ± SEM. * *p* < 0.05, ** *p* < 0.01.

**Figure 4 cells-11-03186-f004:**
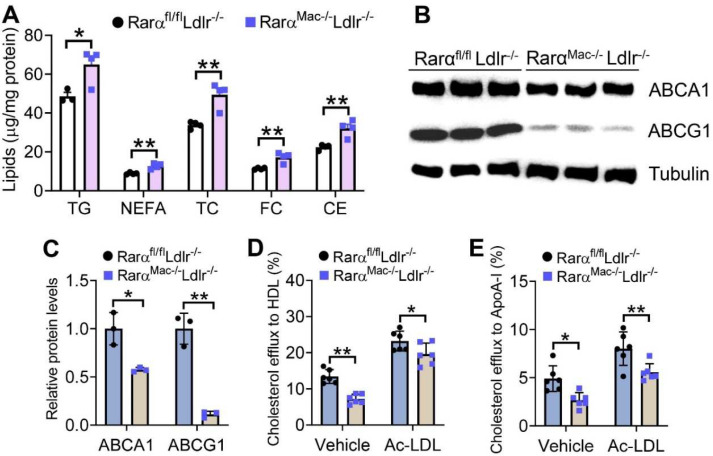
Loss of RARα in Western diet-fed *Ldlr^-/-^* mice promotes foam cell formation by suppressing cholesterol efflux to HDL and ApoA-I. *Rarα**^fl/fl^**Ldlr^-/-^* and *Rar**α**^Mac-/-^**Ldlr^-/-^* mice were fed a Western diet for 16 weeks. The levels of intracellular lipids (**A**) and proteins (**B**,**C**) in peritoneal macrophages were determined (*n* = 3–4). Macrophage cholesterol efflux to HDL (**D**) or ApoA-I (**E**) was determined (*n* = 6). In (**C**–**E**), ordinary two-way ANOVA with Tukey’s multiple comparisons test was used for statistical analysis. All the data are expressed as mean ± SEM. * *p* < 0.05, ** *p* < 0.01.

**Figure 5 cells-11-03186-f005:**
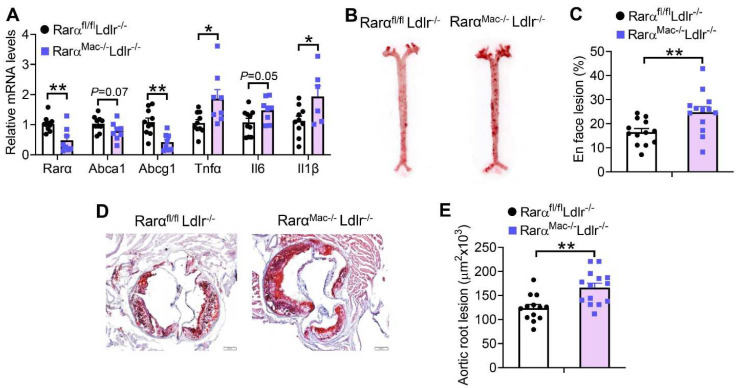
Loss of RARα in macrophages promotes the development of atherosclerosis in *Ldlr^-/-^* mice. *Rar**α**^fl/fl^**Ldlr^-/-^* and *Rar**α**^Mac-/-^**Ldlr^-/-^* mice were fed a Western diet for 16 weeks. The mRNA levels of genes in the plaques of aortic roots were quantified (**A**) (*n* = 8–10). Representative images (**B**) and plaque size (**C**) of *en face* aortas as well as representative images (**D**) and plaque size (**E**) of aortic roots are shown (*n* = 13–16). In (**A**,**C**,**E**), unpaired two-tailed Students’ *t*-test was used for statistical analysis. All the data are expressed as mean ± SEM. In (**D**), scale bar = 1 mm. * *p* < 0.05, ** *p* < 0.01.

## Data Availability

The data are available from Y.Z. (Yanqiao Zhang) upon request.

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
