# Peer review of "Retinoic Acid Receptor Alpha (RARα) in Macrophages Protects from Diet-Induced Atherosclerosis in Mice"

_cells, 2022, doi:10.3390/cells11203186_

Round 1
Reviewer 1 Report
By using a series of in vitro, in vivo, biochemical and molecular biology techniques in WT and RARalpha knockout cells and tissue samples, Bawa and collaborators demonstrated the protective role of RARalpha expression in macrophages in Diet-induced Atherosclerosis in mice. The study is very interesting and deserves further consideration in Cells.
Major points:
1) Why did the authors use male instead of female mice? Would females present a different phenotype in terms of RARalpha activation or deletion?
2) Can the RAR agonists AtRA and AM580 cause protection in vivo?
3) What is the impact of RARalpha deletion and activation in mouse brown and white adipose tissues?
4) Please provide a complete description of qPCR and WB techniques, including details on primer sequences, PCR cycles,, antibodies and their dilutions, incubation periods, etc.
5) Discussion and conclusions must be revisited according with the above points.
Author Response
1) Why did the authors use male instead of female mice? Would females present a different phenotype in terms of RARalpha activation or deletion?
Response: Using macrophages isolated from RARalpha wild-type or knockout mice treated with or without RARalpha agonists for 24 or 48 h, we showed that macrophage RARalpha activation inhibited inflammation and induced cholesterol efflux whereas RARalpha deficiency had opposite effects. Importantly, these effects were also observed in vivo. Based on these observations, we speculate that the phenotypes of male mice versus female mice would be similar in terms of RARalpha activation or deletion. Therefore, we used male mice in the current study. However, more studies are needed to determine whether female mice and male mice have similar phenotypes in vivo (see Discussion, lines 319-322).
2) Can the RAR agonists AtRA and AM580 cause protection in vivo?
Response: Yes, both AtRA and AM580 are shown to protect against diet-induced atherosclerosis or hyperlipidemia in vivo (see Discussion, lines 291-300).
3) What is the impact of RARalpha deletion and activation in mouse brown and white adipose tissues?
Response: AtRA has been shown to promote perivascular adipose tissue browning and adiponectin production in Apoe-/- mice. Adipocyte-specific expression of dominant-negative RARalpha causes glucose intolerance and hepatic steatosis in mice. Nonetheless, the underlying mechanism is not well understood (see Discussion, lines 312-317).
4) Please provide a complete description of qPCR and WB techniques, including details on primer sequences, PCR cycles, antibodies and their dilutions, incubation periods, etc.
Response: The details of RT-qPCR and WB techniques are provided in the methods section of the manuscript (lines 112-150). Primer sequences are provided in Table 1 in Supplementary Information.
5) Discussion and conclusions must be revisited according with the above points.
Response: The above points have been addressed in the Discussion, conclusions, or Methods.
Reviewer 2 Report
The manuscript by Bawa and colleagues nicely demonstrated macrophage RARa promoted ABCA1 and ABCG1 mediated cholesterol efflux in macrophages. Macrophage-specific Rara deficiency exacerbated inflammation and atherogenesis. The paper is very well written and delivered a clear message. Minor concerns need to be addressed:
1. Rara deficiency impaired macrophage RCT. Plasma lipids, however, were not impacted. Can the authors explain or discuss these findings?
2. There should be a brief introduction for AM580.
3. More details of statistical analysis, eg, what post-hoc analysis was used for each of the experiment, should be given.
4. Appropriate format of gene and protein names should be used. See guidelines: https://www.biosciencewriters.com/Guidelines-for-Formatting-Gene-and-Protein-Names.aspx
Author Response
- Rara deficiency impaired macrophage RCT. Plasma lipids, however, were not impacted. Can the authors explain or discuss these findings?
Response: Macrophage cholesterol efflux has little impact on plasma lipid levels as plasma lipid levels are determined mainly by intestinal lipid absorption, hepatic lipid uptake, hepatic VLDL secretion, and plasma triglyceride hydrolysis. However, macrophage RCT plays a key role in removing cholesterol from macrophages, thus protecting against atherosclerosis. We have discussed this in Discussion (see lines 304-305 in Discussion).
- There should be a brief introduction for AM580.
Response: AM580 is a selective RARα agonist, which is included in the revised manuscript (see lines 108-109, 204-205, 297-300).
- More details of statistical analysis, eg, what post-hoc analysis was used for each of the experiments, should be given.
Response: The details of statistical analysis are given for each experiment in the revised manuscript.
- Appropriate format of gene and protein names should be used. See guidelines: https://www.biosciencewriters.com/Guidelines-for-Formatting-Gene-and-Protein-Names.aspx
Response: We have used the standard format of gene and protein names.
Round 2
Reviewer 1 Report
The authors partially responded to the points raised as question 1 remained without an convincing reply and discussion. Is there any evidence on whether RARalpha can influence male and female inflammatory responses differently?
Author Response
We appreciate the reviewer's comments. In Figure 2D and 2F, isolated macrophages were treated with Ac-LDL +/- AtRA or AM580 for 48 h, and then inflammatory gene expression was measured. During the 48-h culture, hormonal effect disappeared and was not a factor for gene expression. However, we found that RARalpha activation inhibited Tnfalpha and Il6 expression whereas RARalpha deficiency induced their expression. Thus, we think RARalpha inhibits inflammatory response independent of gender or sex hormones.
In the Discussion, we also acknowledge the limitation of our studies because only male mice were used.